# The Role of Insulin Resistance in Cancer

**DOI:** 10.3390/curroncol32090477

**Published:** 2025-08-25

**Authors:** Bal Krishna Subedi, Charishma Bhimineni, Shivani Modi, Atousa Jahanshahi, Katherine Quiza, Daniel Bitetto

**Affiliations:** Jefferson Einstein Montgomery Hospital, 559 West Germantown Pike, East Norriton, PA 19403, USA; bal.subedi@jefferson.edu (B.K.S.); cbhimineni@gmail.com (C.B.); shivani.modi@jefferson.edu (S.M.); j.atousa@gmail.com (A.J.); quiza.katherine@mayo.edu (K.Q.)

**Keywords:** insulin resistance, obesity, breast cancer, colorectal cancer, endometrial cancer, liver cancer, pancreatic cancer

## Abstract

Insulin resistance is a condition, common in obesity and type 2 diabetes, where the body’s cells do not respond properly to insulin, a hormone that controls blood sugar. This review explains how this metabolic problem is a significant risk factor for several major cancers, including those of the breast, liver, and endometrium. The high levels of insulin, chronic inflammation, and altered cellular energy usage associated with insulin resistance create an environment that can fuel the growth and spread of tumors. Since insulin resistance is a modifiable condition, understanding this link opens up new strategies for cancer prevention and treatment, emphasizing the importance of lifestyle interventions and metabolic health.

## 1. Introduction

Insulin resistance (IR) is a pathological condition in which the body’s cells become less responsive to the action of insulin, leading to elevated blood insulin levels. Traditionally associated with type 2 diabetes, obesity, and metabolic syndrome, insulin resistance has gained significant attention for its potential role in cancer development and progression. Insulin, along with insulin-like growth factors (IGFs), is involved in regulating cell growth, metabolism, and survival. Given that cancer is fundamentally a disease of dysregulated cell growth, the potential connection between insulin resistance and cancer has become a focal point of research. This review will examine the evidence supporting the link between insulin resistance and cancer, explore the underlying pathophysiology, and discuss the clinical implications for cancer prevention and treatment.

The relationship between metabolic disorders and cancer has been known for over 100 years and represents a significant public health challenge [1,2]. A meta-analysis examining the relationship between T2DM and cancer risk revealed that diabetes was linked to a 10% higher relative risk (RR) of developing cancer (RR: 110; 95% confidence interval [CI]: 1.04–1.17) [3]. For non-sex-specific cancers, the overall cancer risk is estimated to be 1.15–1.17 times higher in individuals with diabetes relative to the general population [4]. The global burden of T2DM continues to rise at an alarming rate, with projections estimating that over 500 million people will be affected by 2030 [5,6]. While the incidence rates of some major cancers have stabilized or declined in recent years due to public health efforts [7,8,9], the increasing prevalence of metabolic disease presents a significant and ongoing public health concern with potential implications for future cancer rates.

## 2. Search Strategy

This narrative review is based on a literature search of the PubMed, MEDLINE, and Google Scholar databases for articles published up to July 2025. The search terms included “insulin resistance and cancer,” “hyperinsulinemia,” “IGF system,” “diabetes and cancer,” “obesity and cancer,” “inflammation,” “metformin,” “mitochondrial dysfunction,” and “Warburg effect,” in combination with specific cancer types such as “breast,” “colorectal,” “pancreatic,” “liver,” and “endometrial.” Foundational experimental studies, large epidemiological cohorts, meta-analyses, and clinical trials were prioritized for inclusion to provide a comprehensive overview of the topic.

## 3. Core Molecular Mechanisms Linking Insulin Resistance and Cancer

The link between IR and cancer is underpinned by several interconnected biological mechanisms that create a systemic environment rich in growth-promoting signals, inflammatory mediators, and metabolic substrates that can initiate and fuel tumor growth. These mechanisms do not operate in isolation but form a vicious cycle where metabolic dysregulation, inflammation, and oncogenic signaling reinforce one another.

### 3.1. The Insulin/IGF System: An Integrated Signaling Axis

The etiology of insulin resistance reflects a complex interplay between genetic and environmental factors, including a sedentary lifestyle and obesogenic diets [10,11,12]. This state leads to sustained hyperinsulinemia, which promotes carcinogenesis through its effects on the insulin/IGF system. The historical view of separate insulin and IGF systems is now considered obsolete; extensive evidence demonstrates an intimately integrated network [13].

This system’s key components include the insulin receptor (IR) and the IGF-1 receptor (IGF-1R). The IR exists in two isoforms: IR-A and IR-B. While IR-B is the primary metabolic isoform in adult tissues, IR-A is the predominant isoform in fetal tissues and is frequently overexpressed in cancer cells [14,15]. Crucially, IR-A not only binds insulin but also binds the potent mitogen IGF-2 with high affinity, making it a primary physiological receptor for IGF-2 [15,16]. The other IGF receptor, the IGF-2/mannose-6-phosphate receptor (IGF-2R), lacks a kinase domain and functions as a scavenger receptor that degrades IGF-2, thereby acting as a tumor suppressor [17].

In the state of IR-driven hyperinsulinemia,
Direct Mitogenic Signaling: Sustained hyperinsulinemia can directly activate the IR and, at high concentrations, the IGF-1R. Upon binding, these receptors activate two major signaling cascades:
○The Phosphatidylinositol 3-kinase (PI3K)/Akt Pathway: This pathway is pivotal, promoting glucose uptake, protein synthesis, and cell growth while strongly inhibiting apoptosis (programmed cell death) [18,19].○The Ras/Mitogen-Activated Protein Kinase (MAPK) Pathway: This pathway primarily drives cell proliferation, migration, and differentiation, contributing to tumor expansion and metastasis [18].Altered IGF-Axis Signaling: Hyperinsulinemia suppresses the hepatic production of IGF-binding proteins (IGFBP-1 and IGFBP-2), which may increase the bioavailability of IGF-1 [20,21]. However, the direct link between circulating IGF-1 levels and cancer risk is a subject of ongoing scientific debate. Bloodstream levels of IGF-1 have not been consistently reproducible as prognostic markers, and the clinical failure of all IGF-1R drug blockers has called into question the role of the IGF-1/IGF-1R axis as a primary driver of malignancy (as reviewed in [22,23]). A more scientifically supported view is that the potent cellular and tissue growth-promoting signals (e.g., via the PI3K and MAPK pathways) induced by hyperinsulinemia contribute to the benign proliferative stages of tumor development. These signals can create a permissive environment that synergizes with other intra-tumoral events necessary for the switch to a malignant phenotype. To date, no definitive evidence suggests that activation of the insulin or IGF-1 receptor by itself can trigger the “benign versus malignant transformation switch.” In contrast, a more direct role in tumor progression has been demonstrated for IGF-II, which is often secreted by cancer cells and acts as a key autocrine/paracrine factor in the tumor microenvironment, promoting the angiogenic switch and malignant progression [24].

### 3.2. Chronic Inflammation and Dysregulated Adipokines

Chronic low-grade inflammation, driven largely by obesity, is a hallmark of insulin resistance and plays a pivotal role in cancer development [24,25]. In this state, visceral adipose tissue produces an excess of pro-inflammatory cytokines, such as tumor necrosis factor-alpha (TNF-α) and interleukin-6 (IL-6), which are consistently elevated in individuals with IR [26,27]. These cytokines activate inflammatory signaling pathways like nuclear factor kappa B (NF-κB), which promotes cell survival and proliferation [25]. This chronic inflammatory environment fosters carcinogenesis by

Generating reactive oxygen species (ROS) that cause DNA damage and genetic instability [28,29].Promoting angiogenesis through factors like VEGF [30].Suppressing anti-tumor immune surveillance [31].

This process is further mediated by adipokines, bioactive molecules secreted from adipose tissue. In obesity, there is a dysregulated secretion of these molecules, characterized by

Elevated Leptin: This pro-inflammatory adipokine promotes cancer cell proliferation, angiogenesis, and metastasis, partly by activating the PI3K/Akt and MAPK signaling pathways [32,33].Reduced Adiponectin: This anti-inflammatory adipokine normally exerts anti-tumorigenic effects by activating AMPK, which inhibits cell proliferation and induces apoptosis. Its reduction in obesity removes this protective mechanism [34,35].

The imbalance between high leptin and low adiponectin, coupled with the systemic inflammation from IR, creates a pro-tumorigenic environment that is highly conducive to cancer initiation and progression [36].

### 3.3. Metabolic Reprogramming, Hypoxia, and Plasticity

Insulin resistance profoundly alters cellular metabolism, creating conditions that support tumorigenesis. Cancer cells exhibit metabolic reprogramming, most famously the Warburg effect, where they preferentially metabolize glucose through aerobic glycolysis rather than more efficient oxidative phosphorylation (OXPHOS). This metabolic shift, while less efficient for ATP production, rapidly generates the biosynthetic precursors (nucleotides, lipids, and amino acids) needed for fast proliferation [18,19]. The hyperglycemia and hyperinsulinemia characteristic of IR provide abundant glucose to fuel this process.

This altered metabolism is intimately linked to hypoxia. The rapid, disorganized growth of tumors often outstrips their blood supply, leading to hypoxic (low-oxygen) regions. Hypoxia activates master regulatory proteins called hypoxia-inducible factors (HIFs), particularly HIF-1α. HIFs reinforce the Warburg effect by upregulating glucose transporters and glycolytic enzymes, and they also promote angiogenesis and metastasis, contributing to a more aggressive cancer phenotype [19,37].

A critical emerging concept is metabolic plasticity. To survive therapeutic stress or nutrient deprivation, cancer cells can adapt their metabolism by shifting to alternative fuel sources. In insulin-resistant states, where circulating free fatty acids are elevated, cancer cells can increase their reliance on fatty acid oxidation. Furthermore, many cancers become dependent on glutamine metabolism to support biosynthesis and maintain redox balance [38]. This adaptability is a key mechanism of treatment resistance. As a result, targeting these metabolic escape routes, for example, with glutaminase inhibitors (e.g., Telaglenastat), has become a promising therapeutic strategy to overcome resistance in metabolically reprogrammed cancers [39,40,41].

### 3.4. Mitochondrial Dysfunction

Mitochondrial dysfunction is a critical mechanism linking insulin resistance and cancer, serving as both a consequence of IR and a driver of carcinogenesis [40,42]. In the insulin-resistant state, impaired cellular metabolism increases the burden on mitochondria, leading to several pro-tumorigenic effects:Increased Oxidative Stress: Dysfunctional mitochondria generate excessive reactive oxygen species (ROS). This chronic oxidative stress damages mitochondrial and nuclear DNA, contributing to the genomic instability that drives cancer initiation [43,44,45].Impaired Apoptosis: Mitochondria are central to programmed cell death (apoptosis). In cancer, this process is often dysregulated by the upregulation of anti-apoptotic proteins like Bcl-2, which prevents the release of cytochrome c and blocks cell death. This not only promotes tumor survival but is also a key mechanism of resistance to many chemotherapies [46,47,48]. Targeting this pathway with BH3 mimetics (e.g., Venetoclax) is a promising strategy to restore apoptotic sensitivity [49].Pro-inflammatory Signaling: Damaged mitochondria release signals that activate inflammatory pathways, such as the NLRP3 inflammasome, further contributing to the chronic inflammatory state that fuels both IR and cancer progression [50,51].

## 4. Insulin Resistance and Specific Cancers

Epidemiological studies consistently link IR with an increased risk of several cancers. A summary of the malignancies with the strongest association is provided in Table 1. The relative risk from a large meta-analysis is detailed in Table 2 [52], and the cancers with the strongest link include

Endometrial Cancer: This cancer has one of the strongest associations with IR and obesity. Hyperinsulinemia directly promotes endometrial cell proliferation and indirectly increases risk by reducing sex hormone-binding globulin (SHBG), which increases the bioavailability of estrogens [53,54].Liver and Pancreatic Cancers: IR is a pivotal factor in the pathogenesis of non-alcoholic fatty liver disease (NAFLD), which can progress to hepatocellular carcinoma (HCC) [55,56]. Chronic hyperinsulinemia is also a major risk factor for pancreatic cancer, potentially by promoting local inflammation and directly stimulating pancreatic cell proliferation [57,58].Colorectal Cancer: High levels of insulin and IGF-1 are associated with an increased risk of colorectal adenomas and carcinomas by stimulating epithelial cell proliferation and inhibiting apoptosis [59,60].Breast Cancer: Particularly in postmenopausal women, IR is linked to increased risk. The key drivers are hyperinsulinemia and increased IGF-1 bioavailability, activating the PI3K/Akt and MAPK pathways, along with interactions with estrogen signaling [61,62,63].

## 5. Therapeutic Strategies and Future Outlook

The mounting evidence linking IR to cancer development has opened new avenues for prevention and treatment strategies. Understanding this relationship provides a clear rationale for targeting metabolic health.

### 5.1. Lifestyle Interventions

Lifestyle modification remains the cornerstone of managing insulin resistance and reducing cancer risk.

Diet: Dietary patterns that are low in refined sugars and processed foods and high in fiber have been shown to significantly improve insulin sensitivity and reduce inflammatory markers [64,65]. Healthy fats, such as those found in the Mediterranean diet, also contribute to these benefits [66].Physical Activity: Regular physical activity, including both aerobic and resistance training, improves insulin sensitivity, reduces visceral fat, and lowers systemic inflammation. A minimum of 150 min of moderate-intensity activity per week is associated with a reduced risk for multiple cancer types [67,68,69].

Interventions that combine both dietary changes and physical activity have been shown to produce superior outcomes for metabolic health [70].

### 5.2. Pharmacological Approaches

Beyond lifestyle, several pharmacological agents that target insulin resistance are under investigation for their role in oncology.
Metformin: This first-line drug for T2DM improves insulin sensitivity, primarily by activating AMP-activated protein kinase (AMPK), which in turn inhibits the pro-growth mTOR pathway [71]. By reducing systemic hyperinsulinemia and inflammation, metformin has shown promise in pre-clinical models and some clinical studies. For instance, one trial found that low-dose metformin could suppress the formation of colorectal aberrant crypt foci, suggesting a potential role in chemoprevention [72]. However, the clinical evidence is mixed, highlighting significant limitations. The large, phase III MA.32 trial, which evaluated metformin as an adjuvant therapy in early-stage breast cancer, found no significant improvement in invasive disease-free survival or overall survival compared to placebo [73]. These mixed results underscore the critical need to identify the specific patient populations and cancer types that are most likely to benefit from metformin therapy.Targeting the insulin/IGF system: Given its central role in driving cell proliferation and survival, the insulin/IGF system is an attractive therapeutic target. Strategies have included IGF-1R monoclonal antibodies (e.g., Cixitumumab) and dual IR/IGF-1R tyrosine kinase inhibitors (e.g., Linsitinib) [74,75]. While promising in preclinical studies, these agents have shown only modest efficacy in clinical trials and face significant challenges, including
▪Metabolic Side Effects: Cross-reactivity with the insulin receptor’s metabolic functions can lead to hyperglycemia, complicating treatment.▪Resistance: Tumors often develop resistance by activating alternative growth factor pathways, limiting the long-term effectiveness of these inhibitors [76].

### 5.3. Future Directions and Emerging Research

To overcome these limitations, future research must focus on a more nuanced and personalized approach.

Predictive Biomarkers: A key priority is the identification of biomarkers to predict which patients will respond to metabolically targeted therapies. This could include circulating markers of IR (e.g., C-peptide and HOMA-IR) or tumor-specific markers like the expression level of the IR-A isoform. Understanding how a patient’s metabolic state influences their response to immune checkpoint inhibitors is another critical area for future investigation.Metabolic Plasticity as a Resistance Mechanism: Cancer cells can evade therapy by adapting their metabolism, for instance by shifting their reliance from glucose to alternative fuel sources like fatty acids or glutamine [38]. This metabolic plasticity is a key mechanism of treatment resistance. As a result, targeting these metabolic escape routes, for example, with glutaminase inhibitors (e.g., Telaglenastat), is a promising strategy to block this adaptation and overcome resistance [39,40].High-Risk Populations: The IR–cancer link has profound implications for specific high-risk groups that warrant further study. This includes investigating the long-term cancer predisposition of newborns of mothers with gestational diabetes, who are exposed to hyperinsulinemia in utero. Similarly, rare tumors that ectopically secrete IGF-2, inducing severe hypoglycemia, represent a unique clinical model of IGF-driven malignancy and can provide valuable mechanistic insights.

## 6. Conclusions

Insulin resistance is a significant and modifiable risk factor for the development and progression of several major cancers. Its oncogenic influence is exerted through a complex and integrated network involving the insulin/IGF system, chronic inflammation, and profound metabolic reprogramming. As the global prevalence of metabolic disease continues to rise, a deeper understanding of this connection is crucial for developing effective prevention and therapeutic strategies. While lifestyle interventions are fundamental, the future of pharmacological treatment lies in personalized medicine. Significant research gaps remain, and addressing them will require a focus on three key areas: robust biomarker discovery to guide patient selection, the development of intelligent combination therapies that can overcome metabolic plasticity, and a better understanding of cancer risk in metabolically vulnerable populations. Successfully navigating these challenges will be critical to reducing the global cancer burden in our increasingly insulin-resistant world.

## Figures and Tables

**Table 1 curroncol-32-00477-t001:** Summary of cancers with a strong association with insulin resistance.

Cancer Type	Key Associated Mechanisms	Strength of Epidemiological Evidence
Endometrial Cancer	Hyperinsulinemia, Increased Bioavailable Estrogens (reduced SHBG), Chronic Inflammation	Very Strong
Liver Cancer (HCC)	NAFLD/NASH Progression, Chronic Inflammation, Direct Mitogenic Effects of Insulin/IGF System	Very Strong
Pancreatic Cancer	Hyperinsulinemia, Local Pancreatic Inflammation, Direct Mitogenic Effects	Strong
Colorectal Cancer	Hyperinsulinemia, IGF-1 Signaling, Chronic Inflammation	Strong
Breast Cancer (Postmenopausal)	Hyperinsulinemia, IGF-1 Signaling, Increased Bioavailable Estrogens, Inflammation	Strong

**Table 2 curroncol-32-00477-t002:** Relative risk of the association between T2DM and cancer. Adapted from “Insulin Resistance and Cancer Risk: An Overview of the Pathogenetic Mechanisms” by Arcidiacono, B. et al., 2012, *Journal of Diabetes Research*, PMC3372318. Licensed under CC BY 3.0. [52].

Cancer	Number (*n*) of Examined Studies	Relative Risk (CI 95%)
Liver	Case–control (*n* = 13)	2.50 (1.80–3.50)
	Cohort (*n* = 7)	2.51 (1.90–3.20)
	Cohort (*n* = 18)	2.01 (1.61–2.51)
Endometrium	Case–control (*n* = 13)	2.22 (1.80–2.74)
	Cohort (*n* = 3)	1.62 (1.21–2.16)
Pancreas	Case–control (*n* = 17)	1.94 (1.53–2.36)
	Cohort (*n* = 19)	1.73 (1.59–1.88)
	Cohort (*n* = 3)	1.80 (1.50–2.10)
	Cohort (*n* = 35)	1.94 (1.66–2.27)
Kidney	Case–control (*n* = 9)	1.42 (1.06–1.91)
Biliary Tract	Case–control (*n* = 8) and cohort (*n* = 13)	1.43 (1.18–1.72)
	Case–control (*n* = 10) and cohort (*n* = 5)	1.60 (1.38–1.87)
Bladder	Case–control (*n* = 7)	1.37 (1.04–1.80)
	Cohort (*n* = 3)	1.43 (1.18–1.74)
Colorectal	Case–control (*n* = 6)	1.36 (1.23–1.50)
	Cohort (*n* = 9)	1.29 (1.16–1.43)
	Case–control + cohort (*n* = 14)	1.38 (1.26–1.51)
Esophagus	Case–control (*n* = 6) and cohort (*n* = 11)	1.30 (1.12–1.50)
Non-Hodgkin’s Lymphoma	Case–control (*n* = 5)	1.12 (0.95–1.31)
	Cohort (*n* = 11)	1.41 (1.07–1.88)
	Cohort (*n* = 10)	1.18 (0.99–1.42)
	Cohort (*n* = 3)	1.79 (1.30–2.47)
Breast	Case–control (*n* = 5)	1.18 (1.05–1.32)
	Cohort (*n* = 15)	1.20 (1.11–1.30)

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
