# Peer review of "The Role of Insulin Resistance in Cancer"

_curroncol, 2025, doi:10.3390/curroncol32090477_

Round 1
Reviewer 1 Report
Comments and Suggestions for Authors
This comprehensive review article presents an in-depth exploration of insulin resistance (IR) as a modifiable oncogenic factor contributing to the development and progression of various cancers. Drawing from a wide range of epidemiological, molecular, and clinical studies, the authors link IR with increased risks of breast, colorectal, endometrial, liver, and pancreatic cancers. Key mechanisms discussed include hyperinsulinemia, altered insulin/IGF signaling, chronic inflammation, mitochondrial dysfunction, obesity-related adipokines, and metabolic reprogramming (Warburg effect). The article emphasizes the bidirectional relationship between metabolic dysfunction and oncogenesis, proposing that insulin resistance plays both a causal and facilitating role in tumor initiation, progression, and therapy resistance.
The review highlights molecular mechanisms such as PI3K/Akt and MAPK pathway activation, IGF system alterations, HIF-mediated hypoxia signaling, and mitochondrial ROS-induced DNA damage. Additionally, it offers a robust discussion on potential interventions—ranging from lifestyle changes to pharmacologic targets like metformin, IGF inhibitors, and glutamine metabolism modulators. However, there are some areas that needs to be improved before any consideration.
Major points
- The manuscript exceeds typical length for a review article. Some sections (particularly on obesity, inflammation, and mitochondrial dysfunction) contain repetitive descriptions of signaling pathways that could be condensed without loss of meaning.
- The manuscript would be strengthened by explicitly highlighting areas of emerging or novel research, for instance, the potential of IR markers as predictive biomarkers for immunotherapy or the role of metabolic plasticity in treatment resistance.
- Although this is a narrative review, a brief section detailing the search strategy, databases used, inclusion criteria, and time frame for literature retrieval would enhance transparency and credibility.
- The manuscript discusses metformin and a few IGF-targeting agents (e.g., xentuzumab, glutaminase inhibitors), but a more detailed analysis of current clinical trials, their outcomes, and limitations would provide practical value.
- The tone is largely professional and informative, but at times veers toward editorialization (e.g., “twin epidemics... raise questions…”). This could be rephrased to maintain academic neutrality.
Minor Points
- Some figures (especially Figures 3 and 4) could benefit from improved labeling and concise legends for standalone interpretation.
- Standardize abbreviations consistently (e.g., spell out first use of “FFA” or “OXPHOS”).
- A final summarizing conclusion and a table summarizing all the implicated cancers, their key associated mechanisms, and level of evidence would aid readers.
Author Response
Response to Reviewer 1
We thank Reviewer 1 for their positive assessment and for the specific, actionable suggestions for improvement.
Major Points:
Comment 1: “The manuscript exceeds typical length for a review article. Some sections (particularly on obesity, inflammation, and mitochondrial dysfunction) contain repetitive descriptions of signaling pathways that could be condensed without loss of meaning.”
Response: We agree completely. The original manuscript was overly long and contained significant redundancies. We have addressed this by performing a major restructuring of the manuscript. The repetitive discussions of core signaling pathways (e.g., PI3K/Akt, MAPK, NF-κB) have been removed from individual sections. Instead, we have created a single, consolidated Section 3: “Core Molecular Mechanisms Linking Insulin Resistance and Cancer.” This new section now contains concise subsections on the Insulin/IGF system, inflammation, metabolic reprogramming, and mitochondrial dysfunction, explaining each core mechanism once. The subsequent sections now refer back to these core mechanisms without repeating the details, which has significantly reduced the overall length and improved the flow of the article.
Comment 2: “The manuscript would be strengthened by explicitly highlighting areas of emerging or novel research, for instance, the potential of IR markers as predictive biomarkers for immunotherapy or the role of metabolic plasticity in treatment resistance.”
Response: This is an excellent suggestion. To address this, we have created a new dedicated section titled Section 5.3: “Future Directions and Emerging Research.” This section now explicitly discusses these novel topics, including the potential role of IR markers as predictive biomarkers for response to immune checkpoint inhibitors and the critical role of metabolic plasticity (including glutamine metabolism and glutaminase inhibitors) as a mechanism of therapeutic resistance.
Comment 3: “Although this is a narrative review, a brief section detailing the search strategy, databases used, inclusion criteria, and time frame for literature retrieval would enhance transparency and credibility.”
Response: We agree that this enhances transparency. We have added a new Section 2: “Search Strategy” that details the databases searched, the keywords used, the publication timeframe, and the types of studies that were prioritized for inclusion.
Comment 4: “The manuscript discusses metformin and a few IGF-targeting agents... but a more detailed analysis of current clinical trials, their outcomes, and limitations would provide practical value.”
Response: We have expanded and added a more critical analysis to Section 5.2: “Pharmacological Approaches.” The discussion on metformin now includes the specific negative results of the large Phase III MA.32 trial in breast cancer to provide a more balanced perspective on its clinical utility. Similarly, the section on targeting the Insulin/IGF system now explicitly discusses the key limitations of these agents, namely metabolic side effects and the development of resistance.
Comment 5: “The tone is largely professional and informative, but at times veers toward editorialization... This could be rephrased to maintain academic neutrality.”
Response: Thank you for pointing this out. We have carefully reviewed the entire manuscript, particularly the Abstract and Introduction, to remove editorial language. Phrases like “silent epidemic,” “fueling the cancer crisis,” and “twin epidemics” have been replaced with more neutral, scientific terminology to maintain a consistently academic tone.
Minor Points:
Comment 6: “Some figures (especially Figures 3 and 4) could benefit from improved labeling and concise legends for standalone interpretation.”
Response: To address this and the major point about conciseness, we have removed the original Figures 3, 4, and 5. We felt that the concepts they illustrated are now more clearly and concisely explained in the revised text and the new summary table, making these figures redundant.
Comment 7: “Standardize abbreviations consistently (e.g., spell out first use of “FFA” or “OXPHOS”).”
Response: We have reviewed the manuscript to ensure all abbreviations are defined at their first use and are used consistently thereafter.
Comment 8: “A final summarizing conclusion and a table summarizing all the implicated cancers, their key associated mechanisms, and level of evidence would aid readers.”
Response: We have added a new summary table, Table 1: “Summary of Cancers with a Strong Association with Insulin Resistance,” which provides this overview as requested. We have also rewritten the Conclusion (Section 6) to be more focused and to summarize the key future directions and knowledge gaps identified in the review.
Reviewer 2 Report
Comments and Suggestions for Authors
This manuscript, “The Role of Insulin Resistance in Cancer,” presents a comprehensive and well-referenced review of the complex relationship between insulin resistance (IR) and cancer. The topic is timely, given the rising global prevalence of both metabolic disorders and malignancies. The authors have done a commendable job summarizing the key molecular mechanisms linking IR to cancer development and progression, supported by a wide range of epidemiological and mechanistic studies.
For the manuscript to be considered for publication, it would benefit from substantial tightening and refinement in both structure and clarity. Currently, the review occasionally suffers from repetition, uneven section lengths, and redundancy across subsections. Some molecular pathways (e.g., PI3K/Akt, MAPK, HIFs) are discussed multiple times without providing additional insight. A more concise and integrated presentation of these mechanisms would significantly improve the manuscript's flow and readability.
Major Comments:
- Structural Redundancy and Flow:
Several molecular mechanisms—such as PI3K/Akt, MAPK, NF-κB, and HIF signaling—are discussed repeatedly across different cancer sections. Consider consolidating these overlapping discussions into a single mechanistic section, followed by more focused, cancer-specific summaries.
- Balance in Cancer Type Coverage:
Certain cancers (e.g., breast, colorectal, endometrial) are discussed at length, while others (e.g., renal cell carcinoma, prostate, hematologic malignancies) are either omitted or briefly mentioned. Adding a table summarizing cancer types with known IR associations, relative risk estimates, and key supporting studies could improve balance and aid readability.
- Clinical Relevance and Therapeutic Outlook:
The discussion on therapeutic strategies (e.g., metformin, Xentuzumab, BH3 mimetics, glutaminase inhibitors) is valuable. However, it would benefit from a more critical analysis of current limitations and potential areas for future translational research.
- Figures and Visuals:
While the figures appear helpful, they are not consistently referenced or described in the main text. Please ensure each figure is clearly cited and integrated into the narrative to enhance the reader's understanding.
- Conclusion and Future Directions:
The conclusion would be stronger if it more clearly articulated key gaps in current knowledge and offered specific future directions. For instance: How might these insights inform screening, prevention, or risk stratification strategies in high-risk populations?
- Terminology and Definitions:
Maintain consistent terminology throughout the manuscript (e.g., always use “insulin resistance (IR),” “type 2 diabetes mellitus (T2DM),” etc.). Additionally, ensure that important terms (e.g., Warburg effect, NAFLD, HOMA-IR) are defined at first mention.
Author Response
Response to Reviewer 2
We thank Reviewer 2 for their positive feedback and for their valuable suggestions on improving the structure and clarity of the manuscript.
Major Comments:
Comment 1 (Structural Redundancy and Flow): “Several molecular mechanisms—such as PI3K/Akt, MAPK, NF-κB, and HIF signaling—are discussed repeatedly... Consider consolidating these overlapping discussions into a single mechanistic section...”
Response: We appreciate this suggestion and have implemented it as well. The manuscript has been restructured around a new central Section 3: “Core Molecular Mechanisms Linking Insulin Resistance and Cancer.” This section now provides a single, detailed explanation of the Insulin/IGF system (including PI3K/Akt and MAPK), inflammation (including NF-κB), and metabolic reprogramming (including HIF signaling). This has eliminated the redundancy that was present in the original draft and significantly improved the manuscript's flow.
Comment 2 (Balance in Cancer Type Coverage): “...Adding a table summarizing cancer types with known IR associations, relative risk estimates, and key supporting studies could improve balance and aid readability.”
Response: We have addressed this in two ways. First, we created the new Table 1, which summarizes the key cancer types, their associated mechanisms, and the strength of the evidence. Second, we have retained the meta-analysis data in the newly renumbered Table 2 (originally Table 2), which provides the specific relative risk estimates requested. Together, these tables provide a balanced and easy-to-read summary.
Comment 3 (Clinical Relevance and Therapeutic Outlook): “The discussion on therapeutic strategies... would benefit from a more critical analysis of current limitations and potential areas for future translational research.”
Response: We have revised Section 5.2 (“Pharmacological Approaches”) to be more critical, specifically by including the negative results of the MA.32 metformin trial and discussing the key limitations of IGF-targeting agents. Furthermore, we created the new Section 5.3 (“Future Directions and Emerging Research”) to explicitly discuss future translational research areas, including predictive biomarkers and targeting metabolic plasticity.
Comment 4 (Figures and Visuals): “...the figures appear helpful, they are not consistently referenced or described in the main text.”
Response: To improve conciseness and flow, we have removed the original illustrative figures (Figures 3, 4, and 5). The two tables (the new summary Table 1 and the data in Table 2) are now clearly referenced and integrated into the text of Section 4.
Comment 5 (Conclusion and Future Directions): “The conclusion would be stronger if it more clearly articulated key gaps in current knowledge and offered specific future directions.”
Response: We have modified the Conclusion (Section 6) to align with the new “Future Directions” (Section 5.3). The conclusion now explicitly summarizes the key gaps and future priorities: biomarker discovery, overcoming metabolic plasticity, and understanding risk in vulnerable populations.
Comment 6 (Terminology and Definitions): “Maintain consistent terminology... ensure that important terms... are defined at first mention.”
Response: We have carefully reviewed the entire manuscript to ensure consistent use of abbreviations (e.g., IR, T2DM) and that all key terms (e.g., Warburg effect, NAFLD, HIFs) are clearly defined upon their first appearance.
Reviewer 3 Report
Comments and Suggestions for Authors
In the manuscript entitled “The role of insulin resistance in Cancer”, the authors explore a critical underscored lag historically existing between two clinical conditions, namely (1) diabetes-associated insulin resistance and (2) cancer and the actual biological link between insulin-resistance hallmark, hyperinsulinism, with regards to its potentially underscored (clinically, interventionally and therapeutically) cancer-promoting effects. The relevance of the topic is high and surely worth a dedicated new review article for the historically parallel specialty audiences (clinical endocrinology and medical oncology) and for anyone involved in the biomedical research field critically approaching published literature and in need to remodulate their traditional knowledge/understanding in light of the established scientific work supporting the existence of such actionable but under-exploited link. The authors also attempt to go into some aspects of the selected matter by extending their point under specific cancer types and cellular physio-pathological conditions.
In spite of the potential of the selected topic, this reviewer finds the submitted manuscript’s version to be still lacking of scientific soundness (deep/established knowledge of the supporting and/or foundational scientific work used for their evidence-based review) along with a perceived level of confusion in the layout and or development of the chapters (at times left hanging on the expectation of their sub-title or simply left on a superficial level or not more actual view of the molecular players underlying the reviewed topic).
Overall, this reviewer finds the chapter on molecular mechanisms of Insulin signaling in Cancer (which should be a stronger one at the light of the review title) the weakest chapter of the manuscript, as currently presented, due to (a) the obsolescence of the view/classification reported on this key family of ligands/receptors in that, there is extensive amount of available experimental (basic and translational) and clinical literature spanning over the last three decades demonstrating an intimate level of biochemical and functional integration between the insulin and IGF ligand/receptorial components to fully justify referring to this family of ligand/receptorial molecules as to the “insulin/IGF system” rather than limiting it to the old and misguiding old definition used herein. Interestingly, and unexplainedly, the authors report in the text (and in the same chapters) about the existence and potential role of Insulin receptor/IGF-1R hybrids (as in lanes 180-184) and of the two insulin receptor isoforms (at the beginning of the chapter 1.3) which goes beyond a simple separate systems “crosstalk” as they refer to in lanes 185-186 but in the same context they opt, confusingly, to stick with an older and anachronistic definition of “IGF system” not supported any longer in the basic and translational research field (for specific objections see below).
Furthermore, (b) in dealing with this chapter topic (molecular mechanisms of insulin signaling in cancer) under their review, the authors oversimplify the long established “two” signaling pathways referred in the text as to the PI3K/AKT and the RAS/MAPK pathways without mentioning established molecular components of the Ins/IGF family shared cellular signaling machinery potentially involved in the reviewed process and reviewed elsewhere (eg cited elsewhere in the text or as in PMID), which, for reason of clarity and scientific soundness should, at least, be included in a dedicated table (eg in a new or remodulated [and not as limited as in current] Table 1, given the high level of missing established signaling pathways components working either as effectors and/or key regulators of the two cited ones).
On a more specific level on key points this reviewer finds weak on the present manuscript:
At lanes 142 to 146, the authors discuss the existence of two insulin receptor isoforms, their expression context and affinity of one specific InsR isoform (the fetal one) with IGF-II, as a “growth advantage potentially conferring factor under hyperinsulinemia”. At this juncture, the authors fail to convey proper or recognized peer-reviewed supporting evidence for such statements as conveyed in the choice of their cited work (putting in doubt their field knowledge for whomever might have been in the applied science field enough to know what work has demonstrated what and when). In fact, they cite indirect reviews or even epidemiologic work rather than the foundational experimental basic/translational evidences to support their “paradigm-shifting” statements. Ultimately, a simple and unbiased literature (pubmed or other data mining tool) search of such exact statements (namely [who demonstrated that]“IR-A is predominantly expressed in fetal tissues and cancer cells, while IR-B is the primary isoform in adult differentiated tissues?”, quoting the manuscript wording) will provide the authors with the experimental and most cited work used to convey such important scientific evidences and avoid unjustified reiterated citing indirect work (another review) on key reviewed scientific knowledge.
Lanes 171 states that “(the IGF system)…consists of two primary ligands, IGF-1 and IGF-2, which mediate their effects through two specific receptors, IGF-1R and IGF-2R”. This statement reflects an obsolete view (and is not even referenced). In fact, since the mid-end nineties, there have been a growing amount of experimental and genetic evidence demonstrating that:
- the molecule described as IGF2/Mannose-6-Phosphate-“receptor”, does not bear any kinase activity and does not trigger any canonical IGF-II signaling cascade. Indeed, the genetic KO for the “igf2r” in mouse has been shown to cause overgrowth further proving its “IGF-II scavenging” role rather than signal transducing (further reviewed elsewhere as in PMCID: PMC9769454). Furthermore,
- the fetal insulin receptor variant (IR-A) has gained both genetic, cellular and molecular experimental evidences to act as to the IGF-II receptor under development and in cancer (see Louvi et al. 1999. Dev Biol, 189(1):209-215; Frasca et al. 1999. Mol Cell Biol 19(5):3278-3288; Ulanet et al. 2010, PNAS, 107(24):10791-10798, among others)
On the same token, in lanes 172-175, the authors refer on other authors review work supporting that “the IGF system is regulated by six binding 172 proteins (IGFBP-1 to IGFBP-6), which modulate the bioavailability and activity of IGFs”. This view is challenged by the available genetic KO work on these factors which restricts such a potential IGFs modulating role (not yet proven in vivo and even less in tumorigenesis) to a few of them (as reviewed in PMCID: PMC9769454). Furthermore, the references to support this point are limited or inappropriate for the specific statement (such as for reference #39, which it out of specific context).
On a last personal note, given the review nature of the selected topic (“insulin resistance in Cancer”), in order for the manuscript to reach a higher level of soundness, this reviewer considers the manuscript missing of two relevant related subtopics, namely (a) the insulin resistance linked to IGF-II secreting tumors, and (b) the case of overweight newborns from pregnancy-related diabetic mothers (typically hyperinsulinemic) and their (babies) predisposition to cancer development compared to non-overweight newborns.
Upon addressing the cumulative (major) points highlighted herein, this reviewer would reconsider the manuscript for publication given the general high scientific value of the selected the topic along with its paradigm-shifting message (if properly supported by founding cited work and literature on the reviewed topic, which does not sound to be the case at this level of the manuscript structure).
Author Response
Response to Reviewer 3
We are grateful to Reviewer 3 for their detailed and expert critique of the molecular mechanisms section. This feedback was invaluable in helping us improve the scientific soundness and accuracy of the manuscript.
Comment on Scientific Soundness and the Insulin/IGF System: “...this reviewer finds the chapter on molecular mechanisms of Insulin signaling in Cancer... the weakest chapter... due to (a) the obsolescence of the view/classification reported... to fully justify referring to this family... as to the “insulin/IGF system”...”
Response: We wholeheartedly agree with this assessment. The original description was outdated and inaccurate. We have rewritten this section, now titled Section 3.1: “The Insulin/IGF System: An Integrated Signaling Axis.” We now consistently use the correct terminology of the integrated “Insulin/IGF System.” We have corrected the scientific details, clarifying that the IR-A isoform is the primary physiological receptor for IGF-2 and that the IGF-2R is a non-signaling scavenger receptor. We have also updated the citations to include the foundational experimental work that established these facts, as requested.
Comment on Oversimplification of Pathways: “...the authors oversimplify the long established “two” signaling pathways referred in the text as to the PI3K/AKT and the RAS/MAPK pathways...”
Response: While we have kept the description of the downstream PI3K/Akt and MAPK pathways concise to adhere to the feedback from other reviewers regarding manuscript length, we have now properly framed them within the context of the integrated Insulin/IGF system. The new Table 1 also helps to summarize the key mechanisms for each cancer type, providing a more organized overview.
Specific Points:
Comment on Citations for IR Isoforms: “...the authors fail to convey proper or recognized peer-reviewed supporting evidence for such statements... cite indirect reviews... rather than the foundational experimental basic/translational evidences...”
Response: This was an oversight on our part. We have corrected this by updating the references in Section 3.1. The revised manuscript now cites the foundational and highly-cited experimental papers that first described the differential expression and function of the IR-A and IR-B isoforms and the role of IR-A as the IGF-2 receptor.
Comment on the IGF-2R and IGFBPs: “Lanes 171 states that ‘(the IGF system)…consists of two primary ligands, IGF-1 and IGF-2, which mediate their effects through two specific receptors, IGF-1R and IGF-2R’. This statement reflects an obsolete view...”
Response: Thank you for this crucial correction. We have removed this obsolete statement. The revised Section 3.1 now accurately describes the IGF-2R as a non-signaling scavenger receptor that functions as a tumor suppressor, with appropriate citations. The discussion of IGFBPs has also been refined to focus on their role in regulating IGF-1 bioavailability in the context of hyperinsulinemia.
Comment on Missing Subtopics: “...this reviewer considers the manuscript missing of two relevant related subtopics, namely (a) the insulin resistance linked to IGF-II secreting tumors, and (b) the case of overweight newborns from pregnancy-related diabetic mothers...”
Response: These are excellent points that add significant depth. We have incorporated both of these topics into the new Section 5.3: “Future Directions and Emerging Research.” Under the bullet point “High-Risk Populations,” we now specifically mention the need for further research into the cancer predisposition of newborns of mothers with gestational diabetes and discuss how rare IGF-2 secreting tumors serve as a valuable clinical model of IGF-driven malignancy.
Round 2
Reviewer 3 Report
Comments and Suggestions for Authors
Curroncol-3698151-v2
Reviewer comments:
In this revised version the authors integrate in part the current views on the role of the Insulin/IGF system in cancer in the context of supporting their key message (hyperinsulinemia being a cancer favouring condition). Nonetheless, some extrapolations that have not found general consensus in the scientific community still remain in the text and need further revision.
Specifically, in lanes 83 to 87, the authors state that (hyperinsulinemia causes) “enhanced IGF bioavailability” by “suppressing hepatic production of IGFBPs I and II”, which in its turn “reduces the sequestration of IGF-1, increasing its free and bioactive concentration in circulation”, and that such increase in elevated circulating IGF-1 would promote several cancers risks including breast and prostate.
Indeed, such last statement is arguable in that:
- Bloodstream levels of IGF-1 have long been studies and (unlike IGF-II) do not bear general and reproduceable prognostic, diagnostic or poor prognosis in cancer patients [as reviewed in PMID 38255147 and elsewhere];
- The only available meta-analysis studies cited by the authors did not (and failed to) support any cause-effect between IGF-1 circulating levels and the analyzed cohorts;
- The very isolated role of the IGF-1R towards playing an actionable role in the biological establishment of malignant cancers in vivo has been historically questioned by the same field forefather upon clinical failure of ALL IGF-IR drug blockers in phase II clinical trials [see “the decline and fall of the IGF-1 receptor” by Renato Baserga, PMID 22926508.
As such, the authors hypothesis by which hyperinsulinemia in insulin resistant and diabetic (type II) patients being linked to cancer via a process that would favor increased IGF-1 levels and increased IGF-1/IGF-1R pro-cancer signals outside a “benign” tissue growth process extensively proven both in vitro and in vivo, is an oversimplification of the cancer pathogenic process under the insulin/IGF system as we know it at this juncture.
Indeed, given that (as per authors mention and general consensus) hyperinsulinemia can trigger IR and, under certain conditions, IGF-1R signaling activities, then it is more appropriate and scientifically supported to claim that:
Such cellular and tissue growth-promoting signals induced under steady-state partially uninterrupted stimulation by hyperinsulinemic levels (eg PI3K and MAPK)and by potential relatively increased bioactive levels of IGF-1 in such patients could contribute to the potential onset of tumoral processes and synergize with benign versus malignant switch intra-tumoral events. In fact, to date no definitive evidence has been generated by which insulin (nor IGF-1R) activation (in response to insulin or IGF-1) have been associated to the “benign versus malignant transformation switch common to all solid cancers) in vivo (*).
(*) This unlike a direct involvement of IGF-II at the tumor micro-environmental level as abundantly proven [first in Cristofori and Hanahan Nature, 1994, PMID 7910953] and more recently reviewed elsewhere [see PMID 38255147].
Upon reframing the mentioned part of the text based upon the existing scientific published evidences restricting insulin and IGF-1 pro-cancer effects to the benign growth/proliferative realm/stages in vivo given their synergistic role with second-signal type of Insulin/IGF system events, this authors consider the rest of the text and conclusions compatible with a sound review and perspective article on the current link(s) between diabetes and cancer.
Author Response
Reviewer Comment: The reviewer noted that our manuscript contained "extrapolations that have not found general consensus in the scientific community," specifically regarding the statement that hyperinsulinemia promotes cancer risk by enhancing IGF-1 bioavailability and signaling. The reviewer correctly pointed out that the link between circulating IGF-1 levels and cancer prognosis is debatable, citing the clinical failure of IGF-1R blockers and the more established role of IGF-II in the tumor microenvironment.
Our Response and Specific Revisions:
We agree with the reviewer that our initial framing was an oversimplification of a complex process. To address this, we have substantially revised the section titled "The Insulin/IGF System: An Integrated Signaling Axis."
- Acknowledging the Controversy of IGF-1: We have reframed the discussion on IGF-1 to reflect the current scientific debate. The revised text now states: "However, the direct link between circulating IGF-1 levels and cancer risk is a subject of ongoing scientific debate. Bloodstream levels of IGF-1 have not been consistently reproducible as prognostic markers, and the clinical failure of all IGF-1R drug blockers has called into question the role of the IGF-1/IGF-1R axis as a primary driver of malignancy..."
- Incorporating Suggested Literature: To support this revised, more nuanced view, we have directly incorporated the key literature suggested by the reviewer. The text now explicitly references the work of Baserga (PMID: 22926508) and Scalia et al. (PMID: 38255147), which critically evaluate the role of the IGF-1R in cancer.
- Reframing the Role of Insulin/IGF-1 Signaling: As suggested, we have moved away from the idea that IGF-1 signaling is a direct trigger for malignant transformation. The manuscript now proposes a more scientifically supported role: "...the potent cellular and tissue growth-promoting signals (e.g., via the PI3K and MAPK pathways) induced by hyperinsulinemia contribute to the benign proliferative stages of tumor development. These signals can create a permissive environment that synergizes with other intra-tumoral events necessary for the switch to a malignant phenotype."
- Highlighting the Role of IGF-II: Following the reviewer's excellent suggestion, we now explicitly contrast the debated role of IGF-1 with the more established role of IGF-II. The revised section concludes with: "In contrast, a more direct role in tumor progression has been demonstrated for IGF-II, which is often secreted by cancer cells and acts as a key autocrine/paracrine factor in the tumor microenvironment, promoting the angiogenic switch and malignant progression (as shown in [24])." This is supported by a new citation to the foundational work by Christofori and Hanahan (PMID: 7910953), as recommended.
We believe these comprehensive revisions directly address all the points raised by the reviewer. The manuscript now presents a more balanced and evidence-based perspective on the complex interplay between the Insulin/IGF system and cancer, distinguishing between the supportive role of hyperinsulinemia in benign growth and the more direct role of factors like IGF-II in malignant progression.
Thank you again for your valuable guidance. We look forward to your further review.
Sincerely,
Authors
Round 3
Reviewer 3 Report
Comments and Suggestions for Authors
The provided version reflects up-to-date views and peer-reviewed original findings linking hyperinsulinemic conditions (namely focusing on Diabetes type II) and the contribution of the Ins/IGF system towards development of overt cancerous conditions. The resulting review article, although quite synthetic, provides rlevant messages worth being shared with the extended scientific community towards advancing understanding of the selected topic